Global change impacts on bird biodiversity in South Asia: potential effects of future land-use and climate change on avian species richness in Pakistan

Khaliq Imran 1 2 imrankhaliq9@hotmail.com
Biber Matthias 3
E. Bowler Diana 4
http://orcid.org/0000-0002-7763-1885 Hof Christian 3
1 Department of Aquatic Ecology, Swiss Federal Institute of Aquatic Science and Technology (EAWAG) , Dubendorf, Zurich , Switzerland
2 Department of Zoology, Government (Defunct) College , Dera Ghazi Khan, Punjab , Pakistan
3 Terrestrial Ecology Research Group, Department for Life Science Systems, School of Life Sciences, Technical University of Munich , Freising, Munich , Germany
4 UK Centre for Ecology & Hydrology Maclean Building , Wallingford, Oxford , United Kingdom
Li Chenxi
Electronic publication date: 2023 Oct 6
Publication date: 2023
Volume: 11
Electronic Location ID: e16212
Received 2023 Feb 20; Accepted 2023 Sep 10
Copyright: © 2023 Khaliq et al.
Copyright year: 2023
Copyright holder: Khaliq et al.
License: This is an open access article distributed under the terms of the Creative Commons Attribution License, which permits unrestricted use, distribution, reproduction and adaptation in any medium and for any purpose provided that it is properly attributed. For attribution, the original author(s), title, publication source (PeerJ) and either DOI or URL of the article must be cited.
License URL: https://creativecommons.org/licenses/by/4.0/

Keywords: Species richness, Birds, Climate change, Land-use, Pakistan

Funding: Academic Exchange Services (DAAD) German Federal Ministry of Foreign Affairs 57523426 and 57609236 Academic Transition Grant by the EAWAG Bavarian State Ministry of Science Bavarian Climate Research Network bayklif (project “mintbio”) This work was supported by the Academic Exchange Services (DAAD) with funds from the German Federal Ministry of Foreign Affairs (project IDs 57523426 and 57609236). Imran Khaliq was supported by the Academic Transition Grant by the EAWAG. Christian Hof and Matthias Biber also received support from the Bavarian State Ministry of Science and the Arts via the Bavarian Climate Research Network bayklif (project “mintbio”). The funders had no role in study design, data collection and analysis, decision to publish, or preparation of the manuscript.

==============================
Evaluating the impact of future changes in land-use and climate on species communities, especially species richness, is one of the most important challenges of current research in ecology and conservation. The impact of environmental changes on species richness depends on its sensitivity (i.e., how strongly a given level of change influences the ecological community) and its exposure (i.e., the amount of change that occurs). To examine the sensitivity, exposure, and potential impact of future environmental conditions on bird communities, we compiled data on bird species richness for Pakistan—a neglected region in macro- or country-scale studies. Since bird species richness strongly varies across seasons due to the seasonal occurrence of migratory species in winter, we compared both wintering (migratory plus resident species) and breeding (resident species only) bird richness. We found breeding and wintering species richness to be sensitive to temperature, precipitation and rainfed cropland by being positively related to these factors. Exposure varied regionally, with projected temperature changes being most profound in northern regions while the strongest projected precipitation changes occurred in central and southern regions. The projected impact of future environmental change were highly heterogeneous across the country and differed between the wintering and breeding communities. Overall, the most negatively impacted region was projected to be the Khyber Pakhtunkha province in the North of Pakistan, due to reductions in precipitation and rainfed cropland, resulting in a projected negative impact, especially on wintering species richness. By highlighting the regional and seasonal bird communities most at risk, our findings provide useful information for policy makers to help devise new policies for mitigating negative impacts of future environmental changes on birds within Pakistan.

Introduction

Climate and land-use are the two primary environmental drivers that shape the spatial patterns of biodiversity, including species richness (Antão et al., 2020; Barreto et al., 2021; Evans, Warren & Gaston, 2005; Rahbek & Graves, 2001). Understanding the impacts of these environmental drivers on biodiversity is crucial, as their effects can vary depending on the intrinsic sensitivity of species within local communities (Antão et al., 2020; Foden et al., 2013) and the spatial heterogeneity in the intensity of, or exposure to, these drivers (Udy et al., 2021).

Among several hypotheses explaining the role of climatic factors in spatial species richness patterns, the species-energy hypothesis is particularly prominent. This hypothesis predicts that areas with higher productivity, warmer temperatures, and greater humidity harbour higher numbers of species (Wright, 1983). Such areas can support either faster speciation rates or a higher carrying capacity, allowing for the coexistence of a greater diversity of species (Wright, 1983). The species-energy hypothesis is supported by empirical evidence showing a positive association between temperature, precipitation, and species richness, for instance, explaining over 60% of the variation in species richness at large spatial scales (Hawkins et al., 2003).

Recent changes in climatic conditions may, however, be too rapid for species to adequately respond (Devictor et al., 2008), potentially weakening the environment-diversity relationships (Barreto et al., 2021; Hawkins et al., 2003). Despite this, it is reasonable to expect that many species are undergoing range shifts to track their preferred climatic conditions (Pecl et al., 2017), leading to the continued influence of changing climatic conditions on local and regional species richness. It is worth noting that temperature and precipitation exhibit both spatial and temporal variation (Vichot-Llano et al., 2021), meaning both spatial and temporal axes are crucial when assessing future impacts of climatic changes on species richness.

In addition to climate, human activities have direct effects on biodiversity patterns, primarily through land-use change, which can further modify environment-diversity relationships (Hof et al., 2018; Newbold, 2018; Newbold et al., 2015). Changes in land-use, such as deforestation or urbanization, can render climatically suitable regions less suitable for species (Comte et al., 2021; Pecl et al., 2017). Similarly, changes in water availability in rivers, resulting from climate change, can have an impact on species richness of birds (Li et al., 2013), particularly in arid regions where species tend to aggregate around water bodies (Khaliq et al., 2019). Further, as the human population continues to grow, human population becomes an increasingly important factor affecting biodiversity, leading to competition for resources and space (Newbold et al., 2015). Overall, these different facets of on-going anthropogenic activities may have an increasingly negative impact on species richness patterns.

In light of these factors, it becomes crucial to study the potential responses of wintering and breeding bird species in Pakistan to environmental changes. The region presents an interesting case due to the rapid loss of glaciers, which supply water to the Indus River, the main flyway for wintering species in the area (Kargel, Leonard & Bishop, 2014). The Indus River plays a crucial role in providing a suitable habitat and sustenance for these birds. However, with the rapid loss of glaciers in Pakistan, the availability and timing of water flow in the Indus River may be significantly altered (Kääb et al., 2012). This change could have substantial implications for migratory patterns and wintering ecology of bird species that depend on this flyway. Wintering species may have different resource requirements and seasonal occupancy patterns compared to breeding species, making them particularly exposed to different environmental conditions (Gómez et al., 2016; La Sorte & Jetz, 2012; Zurell et al., 2018). Consequently, it is plausible that wintering and breeding bird species richness show different environment-diversity relationships (Somveille, Manica & Rodrigues, 2019).

The other important environmental factor for birds is increased competition for resources with the human population, especially along rivers (Pakistan is 6th most populous country in the world, https://worldpopulationreview.com/). In recent decades, humans have transformed the natural riverine forest and landscape close to rivers into agricultural land and urban settlements (Sterling, Ducharne & Polcher, 2013) and that has created competition between land for biodiversity and land for local human populations and associated activities. While there have been studies on bird diversity patterns at the local level (Khaliq et al., 2011, 2019; Khan et al., 2008; Mushtaq-ul-Hassan et al., 2011; Umar et al., 2018), we have little information about bird diversity patterns across the entire country of Pakistan. Hence, the relative importance of climate versus land use for shaping present-day and future species richness patterns in Pakistan remains unclear.

In this study, we aim to assess the impact of different drivers on the spatial patterns of bird species richness in Pakistan. We use the vulnerability framework of Foden et al. (2013) and Williams et al. (2008) which decomposes impact into sensitivity (i.e., how much species are affected by a given change) and exposure (i.e., how much the environment is project to change). We first examine the relationship between temperature, precipitation, water discharge from rivers, and land-use (including pastures, rainfed cropland, irrigated cropland, and human population density) and the present-day bird species richness to assess the sensitivity of local bird communities to these environmental drivers. Subsequently, we compare current and future driver intensities to quantify exposure level of species richness to these drivers of change. Finally, we estimate the combined impact of the drivers on species richness by integrating both the quantified sensitivity and predicted exposure. Based on the predictions of the species-energy hypothesis, we expect that species richness is highest in regions with the highest temperature and amounts of precipitation. To explore potential seasonal differences in diversity-climate relationships, we compare the sensitivity and exposure for wintering and breeding bird communities separately. We expect that changes along the Indus River, particularly the growing nearby human population, will have especially large impacts on wintering bird richness due to considerable number of species are migratory. Our findings may have implications for spatial conservation planning in Pakistan, providing insights into the regions most vulnerable to future environmental changes and where declines in species richness could be most severe in the near future.

Materials and Methods

Study area

Pakistan is a mainly latitudinally distributed country in South Asia with very strong latitudinal (and altitudinal) gradients in multiple environmental variables (Fig. S1). The northern parts of the country have some of the world’s highest peaks and mountain ranges with high levels of precipitation and a comparatively high vegetation coverage. The country is dissected into two main halves longitudinally by the Indus River (Kanwal et al., 2015; van Steenbergen et al., 2015). On its western side, there are mountain ranges running parallel to the Indus River and along the entire length of the country. However, the average height of these mountains gradually decreases from north to south. The eastern side of the Indus is a flat plain and has vast areas of agricultural lands. The natural seepage of water from the Indus is towards the eastern side resulting in repeated floods in these areas (i.e., in the Punjab and Sindh provinces). Therefore, the land is rich with alluvial soil and excellent for agricultural practices. The density of the human population is also highest in these regions (Fig. S1, Kanwal et al., 2015; van Steenbergen et al., 2015). Depending on the regions we can loosely define four seasons i.e., winter, spring, summer and autumn, in Pakistan as follows. Winter lasts from October to January. Spring starts in mid-February and lasts until mid-March. Summer season is a bit longer and starts from mid-march and lasts until August. Autumn is from September to mid-October.

Species data

We extracted species ranges for birds occurring in Pakistan from BirdLife (BirdLife International and NatureServe, 2014; http://datazone.birdlife.org/species/requestdis) and gridded each species’ polygons to a raster layer of 0.5 × 0.5° resolution in WGS84 projection resulting in 327 grid cells in total. We only considered range polygons of native species and with extant or probably extant presence. These range maps reflect the best available knowledge of species distributions over the past approximately 30 years. We differentiated between breeding and wintering ranges for each species, where polygons with resident and breeding season were classified as breeding ranges and polygons with resident and non-breeding were classified as wintering ranges. We define wintering birds as all migrant and resident birds, while breeding birds are solely all resident species. To our knowledge resident species do not make seasonal movements within Pakistan. Also, there are no species that breed in Pakistan and leave in the winter. Hence, migratory birds only add to the wintering bird communities. All raster layers were cropped to the extent of Pakistan and masked by the country shapefile obtained from the Database of Global Administrative Areas (https://gadm.org). We then overlaid the resulting raster layers of all species and calculated the number of species in each grid cell for breeding and wintering birds, separately (Fig. 1).

Figure 1 Species richness patterns of breeding and wintering birds in Pakistan.

Species richness (A and B) is calculated by overlaying the species distribution range maps and counting the number of species present in each grid cell of a 0.5 × 0.5° resolution.

Climate and land-use data

We used the EartH2Observe, WFDEI and ERA-Interim data Merged and Bias-corrected for ISIMIP (EWEMBI; Lange, 2019; https://www.isimip.org) dataset for current climatic conditions, by calculating the average conditions in mean seasonal temperature and total seasonal precipitation within a 30 year time frame (1980–2010) separately for summer (March, April, May, breeding months) and winter (October, November, December, migratory months). We selected these seasons for our analysis since we wished to relate climate to each seasonal bird community, where summer climate is relevant for the breeding birds and winter climate is relevant for the wintering birds. Furthermore, we specifically examined the mean values for temperature and precipitation over a three-month period corresponding to each respective season. This approach was chosen because these average values capture the prevailing climatic conditions more effectively, making them ecologically significant and relevant for our analysis. For future climatic conditions, we used the bias-corrected global climate projections from ISIMIP2b for two Representative Concentration Pathways (RCPs; RCP2.6 and RCP6.0), and four global climate models (GCMs; MIROC5, GFDL-ESM2M, HadGEM2-ES and IPSL-CM5A-LR), and again calculated the average conditions over a 30 years’ time period (2035–2065). For both current and future land-use, human population density and water discharge, we used ISIMIP2b simulations using the same RCPs, GCMs and time periods as for the climatic conditions. The land-use simulations are based upon the assumptions of population growth and economic development under the Shared Socioeconomic Pathway SSP2, i.e., a middle-of-the-road development with an objective to cover the middle ground in terms of mitigation and adaptation challenges (Fricko et al., 2017), and the future changes in climatic conditions provided by ISIMIP2b (Frieler et al., 2017). The land-use simulations provide the proportional cover of rainfed and irrigated cropland as well as pastures per grid cell. Annual gridded human population density was also obtained from ISIMIP2b (Frieler et al., 2017), while maximum annual water discharge was provided by Stacke & Hagemann (2012) as part of the ISIMIP2b model simulations using three out of the four GCMs (IPSL-CM5A-LR, GFDL-ESM2M, MIROC5). The maximum annual water discharge in rivers was calculated and again averaged over the two corresponding 30-yr time periods separately for summer and winter conditions. All climate, land-use, human population and maximum water discharge layers were available on a 0.5 × 0.5° spatial resolution in WGS84 (see ISIMIP2b, Frieler et al., 2017).

Statistical analysis

Sensitivity

We quantified the sensitivity of breeding and wintering bird species richness to different environmental factors using generalised least square (GLS) models. We included means of temperatures of respective season, respective total seasonal precipitation, maximum water discharge of the rivers and the proportion of irrigated cropland, rainfed cropland, pastures and human population as predictor variables and species richness as the response variable. We used summer climate data for the breeding bird models and winter climate data for the wintering bird models. We also tested for all combinations of interactions between mean seasonal temperature, total seasonal precipitations and rainfed cropland. However, we conducted separate models for wintering species and resident species. This decision was made to facilitate interpretation as our goal was to understand the factors influencing the richness patterns of each group. We chose not to include more complex interaction terms that would be challenging to interpret. Prior to running the models, we checked for any strong correlations among the set of candidate predictors. For running GLS models, we used the “gls” function in the “nlme” package of R (Pinheiro et al., 2013). We included the geographic coordinates of each grid cell in the error structure (corExp) to account for spatial autocorrelation (Dormann et al., 2007). For further analyses, we only selected those environmental factors that were statistically significant.

Exposure

We estimated the change in environmental drivers in Pakistan during 1980–2010 using mean seasonal temperatures, total seasonal precipitation and rainfed cropland. To quantify future exposure, we calculated the difference between the mean of current conditions (1980–2010) and the mean of future conditions (2035–2065) in each grid cell for each variable. We calculated exposure for wintering birds using climate during winter and for breeding birds using climate during summer. To calculate the significant spatial association of different environmental drivers, we applied a modified t-test using the “t.test” function in the SpatialPack R package (Vallejos & Osorio, 2020).

Impact

To calculate the potential impact of future environmental changes on breeding and wintering species richness, we integrated the estimated sensitivity and exposure of change in each environmental variable. The exposure values represent the magnitude of change in natural units, such as °C, mm per unit area etc. for each respective variable. Sensitivity was measured as the species richness change per unit of change (e.g., per °C) in the respective variable, derived from the coefficients obtained in the GLS model. To calculate the impact for each variable, we multiplied the estimated exposure of change by the corresponding sensitivity estimate, which gives a measure of the impact of each environmental variable in terms of total species richness change due to the total amount of projected change in that variable. If significant interactions were identified between variables, we also included the corresponding interaction coefficients in the impact calculation. These interaction coefficients quantified the additional change in species richness. For the variables with identified interactions, we also multiplied the exposure values of the interacting variables together by their respective interaction coefficients. This allowed us to capture the combined impact resulting from the interaction between those variables. Finally, we summed up all the individual impacts (main effects and interaction effects) to obtain a single value representing the overall impact of future environmental change on breeding and wintering species richness for each specific grid cell. By considering both the main and interaction effects, this approach provides a comprehensive assessment of the potential impacts and helps capture the complexity of the relationships between the environmental variables and species richness.

Finally, we visualized the predicted change in species richness in each grid cell. We used R version 4.0.2 (R Core Team, 2020) for all the analyses.

Results

Species richness pattern and sensitivity

Present-day species richness patterns contrasted for breeding and wintering birds. The highest number of breeding species (n = 280) was found in the Northern Areas. However, the highest number of wintering species (n = 312) was found along the Indus River, suggesting wintering species, many of whom are migrants, are more dependent on water than breeding species (Fig. 1). There was a mean species overlap of 45% between wintering and breeding birds indicating that 55% species of local wintering species are typically migrants.

Species richness was positively associated with temperature, precipitation and rainfed cropland for breeding birds and for wintering birds (Table 1). Mean seasonal temperature (°C) had a positive association with breeding species richness and wintering species richness (Table 1). The results were similar when using the maximum seasonal temperatures (Table S1). Total seasonal precipitation (mm) had a positive association for both breeding bird species richness and for wintering bird species richness (Table 1). Similarly, rainfed cropland (%) had a positive coefficient for breeding bird species richness and also a positive coefficient for wintering bird species richness (Table 1). Additionally, the combined effect of temperature and precipitation on bird species richness was also positive (Table 1), indicating that the positive effects of temperature are greater when there is also more precipitation.

Table 1 Association of species richness of breeding and wintering birds of Pakistan with climate, land-use change, water discharge in rivers, and human population density.

Species richness of breeding birds and wintering birds was modeled as a function of mean seasonal temperature, total seasonal precipitation, maximum water discharge, pastures, rainfed cropland, irrigated cropland and human population in a generalized least squares model while adding the spatial covariance structure in the variance-covariance matrix. We also added interaction terms between mean seasonal temperature, total seasonal precipitation and rainfed cropland. All variables are averaged over 0.5 × 0.5° resolution. Variables were kept in their original units for interpretation. All significant values are shown in bold.

Breeding birds	
	β	Std. error	t value	P-value	
Mean seasonal temperature (°C)	9.4133	±2.74	3.42	<0.001	
Total seasonal precipitation (mm)	15.18	±2.95	5.14	<0.001	
Maximum water discharge from rivers (m3s−1)	−0.10	±0.43	−0.23	0.81	
Pastures (%)	0.68	±0.77	0.88	0.37	
Rainfed cropland (%)	4.72	±1.09	4.30	<0.001	
Irrigated cropland (%)	0.73	±1.08	0.68	0.49	
Human population	−0.100	±0.82	−0.12	0.85	
Temperature:precipitation	4.04	1.42	2.83	0.004	
Temperature:rainfed cropland	−4.66	1.54	−3.02	0.002	
Precipitation:rainfed cropland	−1.14	0.96	−1.18	0.23	
Temperature:precipitation:rainfed cropland	1.03	0.83	1.24	0.21	
Wintering birds	
	β	Std. error	t value	P-value	
Mean seasonal temperature (°C)	20.26	±0.38	6.86	<0.001	
Total seasonal precipitation (mm)	11.19	±0.06	3.95	<0.001	
Maximum water discharge from rivers (m3s−1)	0.27	±0.00001	0.50	0.61	
Pastures (%)	0.17	±15.49	0.13	0.89	
Rainfed cropland (%)	4.06	±8.63	3.12	0.001	
Irrigated cropland (%)	0.75	±4.95	0.70	0.48	
Human population	0.01	±0.000001	0.21	0.83	
Temperature:precipitation	3.47	1.42	2.45	0.01	
Temperature:rainfed cropland	−2.65	1.66	−1.59	0.11	
Precipitation:rainfed cropland	−0.63	0.92	−0.68	0.49	
Temperature:precipitation:rainfed cropland	1.81	0.95	1.90	0.05	

However, the variables representing maximum water discharge from rivers, pastures (%), irrigated cropland (%), and human population did not show significant associations with breeding bird species richness (p > 0.05, Table 1). Overall, the results demonstrate significant associations between certain environmental variables and both breeding and wintering bird species richness, highlighting the importance of temperature, precipitation, and land-use characteristics in shaping avian communities.

Exposure

In the future, the highest increases in summer and winter temperatures will be expected in the Northern Areas with mean seasonal temperatures projected to rise by 2.5 °C compared to current temperatures for both summer and winter seasons under a mild warming scenario (RCP2.6; Figs. 2A and 2B ). Across grid cells, mean change in winter temperature (temperature =1.13 °C) was larger than the mean change in summer temperature (temperature = 0.95 °C). Similarly, the largest reduction in precipitation (30 mm) is projected to occur in central Pakistan during the summer season and in Khyber Pakhtunkha during the winter season (Figs. 2C and 2D ). The largest reduction in rainfed cropland is projected along the Indus River (Fig. 2E).

Figure 2 Future exposure of significant environmental drivers of species richness.

(A) Summer temperature, (B) winter temperature, (C) summer precipitation, (D) winter precipitation, (E) rainfed cropland. Exposure to mean seasonal temperature, total seasonal precipitation and rainfed cropland is calculated as the difference between the mean of current (1980–2010) and future conditions (2035–2065) of each variable per grid cell. For temperature and precipitation, we calculated the exposure for the future scenario RCP2.6, representing a relatively mild level for future global warming, and as an average across four different GCMs. (For additional RCPs and a separate depiction of different GCMs, see Figs. S2–S4 in the Supplemental Files). Future data for cropland was only available per annum and not separately for summer and winter.

Under projected future environmental conditions, wintering birds will be most exposed to winter temperatures and rainfed cropland changes in the Northern Areas and in the Khyber Pakhtunkha province, which are species rich. By contrast, breeding birds may become most exposed to summer precipitation change in the regions of Northern Areas and Khyber Pakhtunkha (Fig. 2). Overall, the results across two representative concentration pathways and across four global climate models are consistent with the results presented in the main text for scenario RCP2.6 (Figs. S2–S4).

Future potential impact

Our findings reveal differential impacts on bird species richness in Pakistan, with a greater positive projected combined impact of temperature, precipitation and rainfed cropland change on breeding compared to wintering bird species richness (Fig. 3). The average change in the breeding bird richness across the grid cells due to projected environmental change was 2.39 (interquartile range of impacts across grids = −74, +110). By contrast, the average change in the wintering bird richness across grid cells due to projected environmental change was −9 (interquartile range of impacts across grid cells = −34, +25). Since we found that all environmental variables were positively related to richness, only declines in the values of the environmental variables could cause losses of species, which were only observed with cropland and precipitation. By comparing Figs. 2 and 3, we can identify where changes in the environmental variables match the regions of change of bird richness. Declines in precipitation in central Pakistan explain the greatest projected loss of breeding bird species in this region. While declines in precipitation in Khyber Pakhtunkha and Balouchistan and rainfed cropland in Northeast Pakistan explain the greatest projected losses of wintering bird species in these regions. By contrast, increases in breeding bird richness in the regions of upper Punjab and Khyber Pakhtunkha can be explained by an increase in summer precipitation. Similarly, increase in wintering bird richness in Western Khyber Pakhtunkha can be explained by an increase in winter precipitation.

Figure 3 Potential combined impact of the projected changes of the significant drivers (precipitation, temperature and rainfed cropland) on breeding and wintering species richness in Pakistan.

The impact values represent the predicted change in species richness due to projected change in the environmental conditions, calculated by integrating estimates of sensitivity (Table 1) and projected environmental change between current (1980–2010) and future conditions (2035–2065) in each grid cell with a 0.5 × 0.5° resolution (Fig. 2). (A) Combined impact, breeding birds; (B) combined impact, wintering birds.

However, when we considered future temperature change separately, impacts were projected to be highest in the Northern Areas with a higher bird species richness for wintering than for breeding birds (Figs. 2, S3 and S4).

Geographically, the combined future effects of temperature, precipitation, and rainfed cropland exhibit the highest projected impacts in the Northern Areas, characterized by a substantial increase in breeding and a notable decrease in wintering bird richness (Fig. 3). These results remained consistent across the two representative concentration pathways and four global climate models, further supporting the findings presented in the main text (Fig. S5).

Discussion

Pakistan is among the countries where conservation research is largely under-represented (Wilson et al., 2016). To fill this knowledge gap, we evaluated the sensitivity, exposure and future potential impact of changing environmental conditions on bird species richness. We found that breeding and wintering species richness are sensitive to both climate (temperature and precipitation) and rainfed cropland. The largest observed changes in temperature, precipitation and rainfed cropland were recorded in the species-rich areas of northern Pakistan, Khyber Pakhtunkha and along the Indus River. The future potential impacts of different drivers are projected to vary geographically. The combined impact of temperature, precipitation, and rainfed cropland is significantly positive for breeding birds in Khyber Pakhtunkha and Northern Areas. However, for wintering birds in the same regions, the combined impact is significantly negative. Overall, we identify Khyber Pakhtunkha as a hotspot of both future environmental drivers and of bird diversity for both breeding and wintering birds; hence, this area may need to be especially considered in spatial conservation planning.

Temperature, being related to energy and productivity, was a strong driver of species richness of both breeding and wintering birds in Pakistan. Our results are consistent with other studies, where energy-related variables have been found to be strong predictors of species richness of birds, mammals, amphibians, reptiles as well as plants (Carrara & Vázquez, 2010; Connell & Orias, 1964). Our results suggest that bird species richness increases with increasing mean temperatures. Similar positive relationships have been reported in neighbouring countries including China and elsewhere (Luo et al., 2012; Waldock, Dornelas & Bates, 2018).

Bird species richness is higher in high-energy areas, such as Khyber Pakhtunkha and upper Punjab for both breeding and wintering birds. Elevated temperatures in the future are expected to have a higher positive impact on wintering compared to breeding bird richness. This could be attributed to the greater magnitude of temperature exposure during the winter months, which potentially reduces the energetic cost for these birds (Jha, 2013; Juliana & Mitchell, 2016). Moreover, higher winter temperatures may result in increased zooplankton abundance through the growth of phytoplankton (Pomati et al., 2020), thereby providing a greater food supply for wintering water birds. Notably, the highest aggregation of wintering birds is observed along rivers, including the Indus River (Frank & Conover, 2021).

The decrease in precipitation levels is especially observed across Pakistan during summer months. For instance, Balouchistan experienced the largest reduction in precipitation and largest increase in temperatures, which may explain the lowest number of breeding and wintering birds in Balouchistan. Monsoon rains have a cascading effect on the whole ecosystem in Balouchistan as summer rains can increase the vegetation growth, which in turn may have a positive impact on wintering birds. However, if there is less or no rain during the summer months, a lack of resources may also drive species away from Balouchistan as it is mostly desert (Liang et al., 2020; Liu et al., 2019). However, it is important to note that significant changes in precipitation and temperatures have been observed in distinct geographical regions for breeding and wintering bird communities. This means that the positive impact of one variable may be counteracted by the negative impact of another variable. For instance, in Balouchistan, the potential positive effect of temperature on bird species richness could be offset by the negative effects resulting from reduced precipitation and rainfed cropland availability. This holds true for both breeding and wintering bird populations.

Rainfed cropland exhibits a positive association with both breeding and wintering bird species richness due to its potential as an excellent foraging habitat for various bird species (Smith et al., 2018). Seasonal crops cultivated in rainfed cropland attract arthropods, which serve as prey for birds (Smith et al., 2018). However, future projections indicate a decline in available rainfed cropland areas, which is expected to have a negative impact on bird richness. The most pronounced negative effect of changes in rainfed cropland on breeding and wintering bird species richness is anticipated along the Indus River. It is noteworthy that there is a natural elevational gradient from west to east along the Indus River, with numerous canals on its eastern bank (Inam et al., 2008; Tariq et al., 2021). The western border of the Indus River comprises more arid land, which is projected to decrease, explaining the amplified negative impact of rainfed cropland in this region. Although the elevation itself does not directly influence species richness, it facilitates water seepage. Therefore, any reduction in precipitation will have adverse consequences for rainfed cropland, subsequently impacting species richness. The combined occurrence of reduced precipitation and diminished rainfed cropland along the Indus River suggests a potential scarcity of resources for birds in the future, which could negatively affect species richness. Lastly, on the flip side any expansion of cropland will result in the reduction of native habitats for several species.

The combined impact of temperature, precipitation and rainfed cropland for wintering birds particularly in Khyber Pakhtunkha and upper Punjab is strongly negative and may be especially at risk because of the of rapid changes in these three drivers and their co-occurrence with high human population density, e.g., Khyber Pakhtunkha and upper Punjab. Here, it can be expected that the future increase in the human population will lead to an increase in different human activities that may negatively impact bird diversity (Newbold, 2018; Newbold et al., 2015). For breeding birds, the combined negative impact of temperature, precipitation and rainfed cropland may become strongest along the Indus River. The differences in combined impact for breeding and wintering birds reveal the importance of seasonal differences in climate. Therefore, it is important to consider seasonal effects when evaluating the climatic vulnerability of birds (Eyres, Böhning-Gaese & Fritz, 2017).

Our study has several limitations. First, our analysis is based upon range maps using species presences only. These expert range maps give a broad overview on species’ distributions, but projections of potential changes in species richness based on them should not be mistaken as precise predictions of changes in species numbers, as they rather represent coarse estimates of regional trends. Furthermore, each species was given equal weight in each grid cell where the species was present. Weighting by species abundance may yield different biodiversity patterns, even though presence data should be sufficient for studying species richness (Chase et al., 2019). Second, we have not investigated the potential impact of future environmental changes on the functional and phylogenetic diversity within communities (Stewart et al., 2022; Voskamp et al., 2022). Factors such as species’ range shifts, habitat alterations, and the potential loss of specialized habitats can significantly influence functional and phylogenetic diversity, potentially resulting in the emergence of more homogenized communities (Davey et al., 2012; Finderup Nielsen et al., 2019). Thirdly, our impact analysis is based upon projected environmental data for the future derived from simulations that come with different components of uncertainty. Thus, the magnitude of impact also needs to be interpreted in light of a certain level of uncertainty; however, the mean trajectory should be robust, and we found similar patterns across different RCPs and GCMs (Figs. S2–S5). Fourth, we assessed bird richness sensitivity using a space-for-time approach; therefore, we cannot rule out that other covariates explain some of the environment-richness relationships. For instance, the impacts of rainfed cropland might rather reflect environmental conditions that affect both birds and where crops are planted. Finally, we assumed simple linear relationships, but threshold effects might mean that some variables change in direction and magnitude of effect as their intensity increases further. This may be especially important for temperature, since negative effects of warming might only become apparent at the highest temperatures.

Conclusions

We quantified the sensitivity of breeding and wintering bird species richness to temperature, precipitation and rainfed cropland in Pakistan. For both breeding and wintering birds, we found species richness to be positively associated with temperature, precipitation and cropland, and we expect that temperature and precipitation will continue to be the main drivers of bird species richness in this area. Reductions in precipitation in areas that currently show high bird species richness, such as Khyber Pakhtunkha, indicate likely negative impacts of future precipitation change. However, humans will continue to cause changes in agricultural landscapes, habitats and water resources, which will add additional pressures on bird species richness. Such overlap between bird and human populations makes bird species richness in Pakistan especially vulnerable to change. Our results highlight the most vulnerable regions of Pakistan for birds, including Khyber Pakhtunkha. Hence, our results are important for policy makers to take steps in mitigating the effects of climate change or conserving the bird fauna of Pakistan.

Supplemental Information

Supplemental Information 1 Distribution of elevation (m) and human population across Pakistan.

The internal borders indicate the borders of the provinces.

Click here for additional data file.

Supplemental Information 2 Exposure of mean seasonal temperature change (summer and winter) for different scenarios and global climate models (GCMs).

Exposure is calculated as the difference between mean of current conditions (1980–2010) and mean of future conditions (2035–2065) in each grid cell of 0.5 × 0.5° resolution. Units are °C.

Click here for additional data file.

Supplemental Information 3 Exposure of total seasonal precipitation change (summer and winter) for different scenarios and global climate models (GCMs).

Exposure is calculated as the difference between mean of current conditions (1980–2010) and mean of future conditions (2035–2065) in each grid cell of 0.5 × 0.5° resolutions. Units are in mm.

Click here for additional data file.

Supplemental Information 4 Exposure of rainfed cropland (summer and winter) for different scenarios and global climate models (GCMs).

Exposure is calculated as the difference between mean of current conditions (1980–2010) and mean of future conditions (2035–2065) in each grid cell of 0.5 × 0.5° resolution. Units are in percentage.

Click here for additional data file.

Supplemental Information 5 Figure showing the combined impact of significant drivers of species richness.

Impact of mean seasonal temperature, total seasonal precipitation and cropland is calculated as the product of slope of each variable in regression model (i.e., species sensitivity) and difference between mean of current conditions (1980–2010) and mean of future conditions (2035–2065) (i.e., species exposure) of each variable in each grid cell of 0.5 × 0.5° resolution. For temperature and precipitation, we calculated the combined impact for future scenario RCP2.6 and RCP6 for four different general circulation models (GCMs). Units are number of species.

Click here for additional data file.

Supplemental Information 6 Breeding birds.

Click here for additional data file.

Supplemental Information 7 Wintering birds.

Click here for additional data file.

Supplemental Information 8 Supplementary Figures.

Exposure of temperature (summer and winter) for different scenarios and global climate models (GCMs). Exposure is calculated as the difference between mean of current conditions (1980–2010) and mean of future conditions (2035–2065) in each grid cell of 50 by 50 equal area grid cell.

Click here for additional data file.

Additional Information and Declarations

Competing Interests

Author Contributions

Data Availability

Christian Hof and Imran Khaliq are Academic Editors for PeerJ.

Imran Khaliq conceived and designed the experiments, performed the experiments, analyzed the data, prepared figures and/or tables, authored or reviewed drafts of the article, and approved the final draft.

Matthias Biber conceived and designed the experiments, performed the experiments, analyzed the data, prepared figures and/or tables, authored or reviewed drafts of the article, and approved the final draft.

Diana E. Bowler conceived and designed the experiments, authored or reviewed drafts of the article, and approved the final draft.

Christian Hof conceived and designed the experiments, authored or reviewed drafts of the article, and approved the final draft.

The following information was supplied regarding data availability:

The raw data is available in the Supplemental Files.

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
