# Peer review of "Global change impacts on bird biodiversity in South Asia: potential effects of future land-use and climate change on avian species richness in Pakistan"

_PeerJ, doi:10.7717/peerj.16212_

## Round 0.1 · original submission · Major Revisions

Reviewers' comments on your work have now been received. The manuscript has been assessed by two reviewers. Reviewers indicated that the abstract, the introduction, the method, the result, and the discussion sections should be improved. Moreover, English editing and academic writing service is needed. I agree with this evaluation and I would, therefore, request for the manuscript to be revised accordingly.

Reviewer 1 ·

Basic reporting

The general story is interesting and could attack general interest in ecology. This study provides implications for environmental changes on birds within Pakistan. The writing is good, but needs improvements.

Experimental design

No comments

Validity of the findings

No comments

Additional comments

I have several comments:
1. The location of River Indus is a little different in Figures 1a and 1b. Is it caused by the season change diverting the river?
2. Figure 2a-d, RCP26 should be RCP2.6.
3. I am curious why the authors do not include 2011-2020 for the current data.
4. Writing needs to be checked carefully to avoid any grammar problems.

Reviewer 2 ·

Basic reporting

The article would greatly benefit from clearly-stated hypotheses and perhaps a more eye-catching title. Even though the English used in the manuscript is mostly ok, I have made a several suggestions on grammar and flow that I believe help guide the reader.

I feel the manuscript lacks "biology": it needs a more in-depth on birds, migration and ecology as a whole to better contextualize their interesting findings. Furthermore, I believe adding pictures of bird species would greatly improve the figure. Additionally, a map of Pakistan with elevation and/or population density would be welcome to illustrate the country's geography.
Raw data was not available.

Experimental design

The manuscript lacks clear hypotheses. These should be added at the end of Introduction to guide the reader.
I have a few issues with the methodology:
i) averaging averages without accounting for variance (and then using these to perform further calculations) is worrisome. These variables (e.g. precipitation, temperature, etc) should be first discussed on how they might affect bird species and thus the metric used: e.g. could median maximum annual temperatures be a more impactful factor than average mean temperatures? Once these are clearly-defined (and calculated) the findings are better supported.
ii) the authors should consider interactions between the variables and/or use a single model with all explanatory variables. With separate models the coefficients are systematically overestimated.

Validity of the findings

Underlying data has not been provided but conclusions are overall discussed both geographically and in terms of patterns. However, I would like to read more in-depth discussion on the ecological processes behind these patterns, especially linking those with species ecology (the authors do it very well when discussing water birds, for example) and with ecological theory (e.g. species-energy theory).

Additional comments

The authors present an interesting work on the diversity patterns of birds in Pakistan, a rich but often overlooked region, and relate these patterns with abiotic factors in order to predict how future changes may impact local and regional richness. The manuscript is well-written (though I have made many minor English corrections) but some sections need clarifying: defining clearly what wintering and breeding species are (this was not especially clear and is paramount to understand the manuscript, if a species can be simultaneously 'breeding' and 'wintering' (e.g. seasonally migrating to a different part of Pakistan) and how are treated species that breed in Pakistan in the summer but leave to winter in other regions (compared to fully resident breeding species that live in Pakistan year-round) . For example, notice how in L210 you mention "Wintering AND migrant" species" (inferring that some are not migrant) but you define wintering species as "all the migratory birds".
Please also note that, added to not clearly defining wintering/breeding species, you often interchange this with 'winter' and 'summer' (see Figure 3) and this is confusing. Furthermore, if the authors choose to make this paper more conservation-oriented, you could do the same analyses but for all species (how is bird richness as a whole impacted by these drivers). Often the whole is greater than the sum of its parts and this make the paper more citable and more useful to policy-makers.

Regarding the analyses, you need to incorporate interactions in your GLS since testing variables separately may result in overestimation of coefficients. Furthermore, I believe your approach is oversimplistic: averaging the averages and using this value without accounting for variance leaves a lot of room for parameters to become spurious and to not reflect reality. Thus you should at least do a sensitivity analysis of how other statistics (median, third quartile, etc) impact your results. Furthermore, you need to justify better your choice of parameters: why is the average mean better than average maximum?

The interpretation of Table 1 is confusing and poorly linked to the table itself. Make it as simple as possible: e.g. "two degrees of temperature species richness increased by about three species faster". Why 2 degrees? Was this the unit of your temperature data? Other example is was a "1x faster increase" but this is basically staying the same. Also notice how you again refer to "variables differed between seasons" even though it should be between bird migratory status.

The whole manuscript (but especially the intro, discussion and figures) would greatly benefit from more biology and more Pakistan: add photos of birds, add a map of Pakistan and dwelve more into species ecology or ecological theory. The manuscript shines when you do so and it makes the paper much more interesting to read. Furthermore, it helps to bring visibility to Pakistan and its fauna, a great merit of your work. Furthermore, I value your acknowledgment of the study limitations, congratulation on discussing them.

Overall, I congratulate the authors on an interesting work on an overlooked region and a topic that is useful to conservation practice. It needs a more elegant analysis to fully support its relevant findings and more well-defined essential aspects such as the difference between wintering and breeding species.

Below I add a few extra line-specific comments:
L41: I believe it would be beneficial to explicitly state the difference between breeding and wintering species.
L119: In the Introduction, please dwelve a bit more into the wintering aspect of birds. Add a few more sentences on migration.

L143: This feels problematic, as its presence has not been confirmed thus could overestimate richness. I would remove or at the least run a sensitivity analyses with and without these species. Alternatively, if it only involves a low number of species, leave it explicit in the text.
L156: The manuscript would benefit from a sentence describing Pakistani seasons.
L183: Which geographical coordinates?
L208: In a grid cell?
L211: What does this overlap refer to?
L222: Please write the full province name.
L289: Please confirm if it is “Balochistan”. If it is referred to as 'Balouchistan' in Pakistan feel free to keep it but be consistent throughout the manuscript.
L306: At the same time, an increase in cropland will decrease native habitat. Please also mention this.
322: Not only that but only on taxonomic diversity. Please briefly mention other unexplored aspects such as functional and phylogenetic diversity and how do you think these could be impacted (e.g. water birds might be closely-related thus phylogenetic diversity may respond to changes).

Annotated reviews are not available for download in order to protect the identity of reviewers who chose to remain anonymous.

---

## Round 0.2 · accepted · Accept

The authors have addressed all of the reviewers' comments.

Reviewer 1 ·

Basic reporting

I think the revision is satisfactory and I recommend accepting the current version. Thanks!

Experimental design

No comment

Validity of the findings

No comment

Additional comments

No comment

Reviewer 2 ·

Basic reporting

The manuscript now has more clear definitions and flow. I still feel that hypotheses should have been more well discussed but they are now briefly included. The authors have also added raw data.

Experimental design

The authors did well to address my concerns regarding their analyses, with sensitivity analyses and justifying why they use some of the metrics. I still think median (since it truly reflects variance) is a better metric that average but the authors now discuss why they use the former in their manuscript.

Validity of the findings

The sensitivity analyses have shown that their results remain valid and the authors have also included interactions between variables.

Additional comments

I still feel that including bird photographs (examples of impaced species) would make your article more eye-catching, as readers relate more to articles in which they can see the actual animals. But I respect the authors' decision and congratulate them on an interesting manuscript that focuses on an overlooked fauna and region.